# METACOGNITIVE SELF-CORRECTION FOR MULTI-AGENT SYSTEM VIA PROTOTYPE-GUIDED NEXT-EXECUTION RECONSTRUCTION

## ABSTRACT

Large Language Model based multi-agent systems (MAS) excel at collaborative problem solving but remain brittle to cascading errors: a single faulty step can propagate across agents and disrupt the trajectory. In this paper, we present MASC, a metacognitive framework that endows MAS with *real-time, unsupervised, step-level error detection* and *self-correction*. MASC rethinks detection as history-conditioned anomaly scoring via two complementary designs: (1) Next-Execution Reconstruction, which predicts the embedding of the next step from the query and interaction history to capture causal consistency, and (2) Prototype-Guided Enhancement, which learns a prototype prior over *normal-step embeddings* and uses it to stabilize reconstruction and anomaly scoring under sparse context (e.g., early steps). When an anomaly step is flagged, MASC triggers a correction agent to revise the acting agent's output before information flows downstream. On the `Who&When` benchmark, MASC consistently outperforms all baselines, improving step-level error detection by up to 8.47% AUC-ROC ; When plugged into diverse MAS frameworks, it delivers consistent end-to-end gains across architectures, confirming that our metacognitive monitoring and targeted correction can mitigate error propagation with minimal overhead.

## 1 INTRODUCTION

Large Language Models (LLMs) have established new frontiers in artificial intelligence, demonstrating remarkable capabilities in few-shot learning, planning, and complex reasoning across a wide range of tasks (Brown et al., 2020; Wei et al., 2022a; Yao et al., 2023b). Building upon these advances, the paradigm has shifted beyond single-agent settings towards LLM-based multi-agent systems (MAS), where teams of intelligent agents collaborate to solve problems. This collaborative approach has proven to be a powerful method for tackling tasks of greater complexity than any single agent could manage alone, achieving impressive results in domains such as scientific discovery (Ghafarollahi & Buehler, 2024; Schmidgall et al., 2025), software engineering (Chan et al.), and strategic decision-making (Huang et al., 2025).

To support and optimize such collaboration, researchers have investigated a wide range of multi-agent communication structures. These include foundational topologies like chains (Wei et al., 2022a; Zhang et al., 2022), trees (Yao et al., 2023a), and stars (Wu et al., 2023a), as well as more complex fully connected or random graphs (Qian et al., 2024). These designs are often tailored to task complexity and communication constraints, aiming to balance performance and efficiency. Notably, recent advances propose learning frameworks that dynamically choose query-conditioned communication topologies (Hao et al., 2023; Liu et al., 2023; Zhang et al., 2024a). Such adaptive systems mark a significant shift from fixed interaction pipelines to more flexible, input-aware collectives, better equipped to exploit the full potential of collaborative agents.

Despite these advancements, the increasing complexity and interconnectivity of MAS introduce a critical vulnerability: their fragility to cascading errors. This is because collaborative structures, while enhancing problem-solving, also act as conduits for error propagation, where the system's success is dictated by its weakest link. Our preliminary study (Section 2.2) reveals that a single agent's error can cause system performance to plummet by over $50\%$. This finding underscores

the urgent need for mechanisms that perform *real-time detection and correction* to maintain the system's operational integrity. However, building such a mechanism faces fundamental challenges. First, obtaining fine-grained, step-level error labels in complex multi-agent interactions is notoriously difficult and costly (Zhang et al., 2025c), rendering standard supervised training impractical. This motivates detectors that learn from abundant normal traces and flag errors as anomalies. Second, error detection is intrinsically context-dependent. Our empirical analysis (Section 2.2) finds that when a step is viewed in isolation, normal and abnormal steps often look alike, so detectors must condition on interaction history. The same analysis also shows that a substantial fraction of errors occur early in the trajectory, when context is scarce, making reliable detection even more difficult.

To address these challenges, we introduce **MASC**, **M**eta**c**ognitive **S**elf-**C**orrection for LLM Multi-Agent Systems that enables online, unsupervised, step-level error detection and self-correction. Our framework contains two novel and critical designs: (1) **Next-Execution Reconstruction**, where the system models the causal dynamics of normal agent interactions by predicting the subsequent step's representation from the historical context. This allows for identifying outputs that violate the learned agent interaction flow. (2) **Prototype-Guided Enhancement**, which learns a stable distributional prior of normal agent behavior to act as a robust reference point. This ensures reliable detection performance even when errors occur early in an execution sequence where historical context is limited. Furthermore, when an anomaly is detected, MASC triggers a correction agent that revises the flagged step before erroneous information propagates downstream, thereby preventing cascading failures. Across the `Who&When` benchmark (Zhang et al., 2025c) and diverse MAS frameworks, MASC improves both detection quality and end-to-end task performance.

- **Formulation.** We formalize step-level error detection for LLM-based MAS as history-conditioned, *unsupervised* anomaly agent detection. This formulation avoiding the need for costly, fine-grained step-level error labels.
- **Framework.** We propose MASC, which combines Next-Execution Reconstruction, a Prototype-Guided prior for stability under sparse context, and an anomaly-triggered self-correction loop for real-time robustness.
- **Experiments.** Our detector achieves up to 8.47% AUC-ROC gains on step-level error detection; as a plug-in to multiple MAS frameworks, MASC delivers consistent accuracy improvements across six benchmarks.

## 2 PRELIMINARIES

In this section, we formalize the structure of LLM-based multi-agent systems and define step-level error detection (Section 2.1); we then analyze the core challenges and the necessity of self-correction (Section 2.2), which motivate our method.

### 2.1 PROBLEM FORMULATION

**Multi-Agent System.** We consider a LLMs-powered multi-agent system $\mathcal{M}$ with a group of $N$ agents, denoted as $\mathcal{N} = \{1, 2, ..N\}$, that operate at discrete time steps. These agents are taking actions in a turn-based protocol, meaning that exactly one agent performs an action at each time step. Formally, the system is described as:

$$\mathcal{M} = \left\langle \mathcal{N}, S, A, P, \phi \right\rangle. \tag{1}$$

Here, $S$ is the set of possible states. $A$ is the global action set; each agent $i \in \mathcal{N}$ can typically perform actions from some subset $A_i \subseteq A$. $\phi(t)$ is a function that indicates which agent is active at time $t$, thus specifying the turn-based rule. $P(s_{t+1} \mid s_t, a_t, \phi(t))$ is the state-transition probability, given that *only one* agent $\phi(t)$ acts at time $t$. $\phi(t)$ is employed to denote the agent that takes an action $a_t$ at time step $t$. A full trajectory $\tau$ can be written as: $\tau = (s_0, a_0, s_1, a_1, \ldots, s_T)$, where $T$ is a terminal time step or when the system enters a terminating state. Based on this formal system, we now specify the problem of error step detection. In this context, we are interested in evaluating each action $a_t$ taken by the corresponding agent $i = \phi(t)$ within the trajectory $\tau$.

**Definition 1 (Error Step Detection in Multi-Agent Systems)** *Given a multi-agent execution trajectory $\tau$, we define an error step as a specific agent-time pair $(i, t)$ indicating that agent $i$ at time*

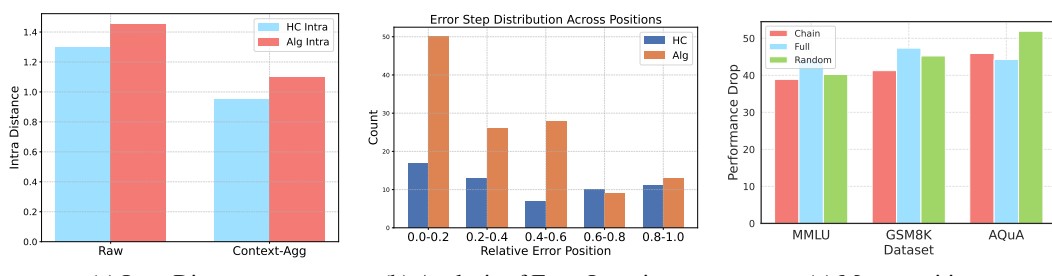

| (a) Intra Distance | (b) Analysis of Error Locations | (c) Metacognitive |

Figure 1: Comparative analysis of intra-distance, error locations, and metacognitive behavior.

*step $t$ performs an incorrect action (e.g., wrong reasoning or decision). The objective of error step detection is, for a given current step $t$ and optionally a set of historical steps $\mathcal{H} = \{(i', t') \mid t' < t\}$, to determine whether the action at step $t$ constitutes an error. Formally, the detection function $\mathcal{D}(i, t, \mathcal{H})$ takes as input the agent $i$, the current time step $t$, and the historical context $\mathcal{H}$, and outputs a binary label:*

$$\mathcal{D}(i, t, \mathcal{H}) = \begin{cases} 1, & \textit{if the action by agent } i \textit{ at time } t \textit{ is erroneous}, \\ 0, & \textit{otherwise}. \end{cases}$$

The primary challenge, which our work addresses, is to construct this detection function $\mathcal{D}$ in a *unsupervised manner*, without access to any labeled error steps during training. This enables the system to monitor itself in real-time and identify deviations from normal, correct execution flows.

## 2.2 PROBLEM ANALYSIS

In this section, we analyze three key challenges that motivate our work: (1) the context-dependent nature of step-level errors; (2) the difficulty of detecting errors early in execution; and (3) the vulnerability of MAS to cascading failures.

**Context-Dependence of Step-Level Errors.** A single step in a multi-agent trajectory is rarely separable from errors without *history*. Using a pretrained BERT encoder, we compare (i) the inter-cluster distance between normal vs. abnormal mean embeddings and (ii) the intra-cluster distance within normal steps. If context were unnecessary, the inter distance would match or exceed the intra distance. Instead, Fig. 1a shows the inter distance is much smaller (e.g., Algorithm-Generated: 0.25 vs. 1.45 in raw embeddings), indicating isolation is insufficient. A simple historical augmentation (nearest neighbor) slightly tightens normal dispersion (e.g., $1.45 \rightarrow 1.10$) but only modestly enlarges the inter gap, underscoring that *choosing the right context is nontrivial*. These findings confirm that step-level anomaly detection in MAS cannot rely on isolated embeddings and must instead exploit contextual and causal dependencies across steps.

**Difficulties of Early-Step Errors.** Beyond the inherent challenge that abnormal steps cannot be directly judged without context, a further difficulty arises when errors occur at early stages of execution, where only limited historical information is available. To quantify this issue, we conduct a statistical analysis on the `Who&When` benchmark, which contains two subsets of multi-agent trajectories: a hand-crafted version (HC) and an algorithm-generated version (Alg). Fig. 1b shows the distribution of error positions relative to trajectory length. We observe that a considerable portion of errors in the Alg subset appear within the first 20% of steps, while errors in the HC subset are more evenly distributed across positions. This evidence highlights that early-step errors are common and can leave detectors with insufficient context, thereby motivating the introduction of a prototype-guided mechanism to provide a stable reference representation when history alone is inadequate.

**Lack of Metacognitive Error Awareness in MAS.** While collaboration aims to improve robustness, current MAS lack metacognitive capabilities to recognize and mitigate their own reasoning failures. We test this via controlled fault injection across three canonical topologies: *chain*, *fully-connected*, and *random*. In each setting, we randomly select one agent and launch a prompt-based attack that forces a misleading output, simulating realistic erroneous agents. Fig. 1c shows the resulting degradation on MMLU, GSM8K, and AQuA: performance drops markedly across all datasets and topologies (e.g., up to 51.9 points on AQuA under the random topology). Thus, collaborative structures, even with designated critic roles, cannot prevent error propagation once an agent fails, highlighting the need for explicit *error detection* and *correction* to guard against cascading failures.

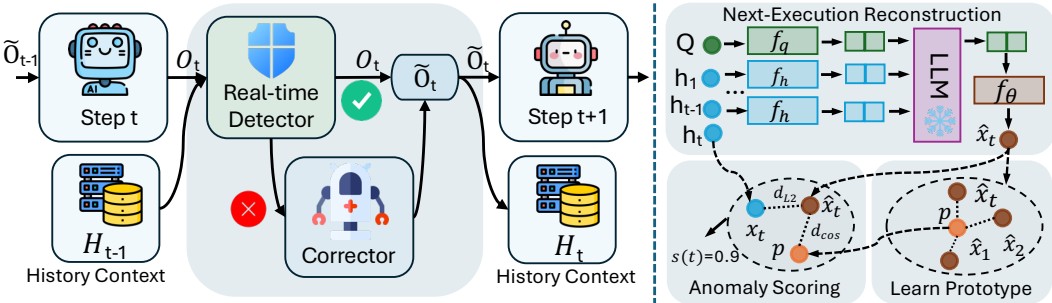

Figure 2: Overview of MASC. Left: At step $t$, the agent's output $\mathcal{O}_t$ and history context $\mathcal{H}_{t-1}$ are sent to a real-time detector; if normal, it passes through, otherwise a correction produces $\tilde{\mathcal{O}}_t$, updates $\mathcal{H}_t$, and is used at $t+1$. Right: Next-Execution Reconstruction takes projected query $\mathcal{Q}$ and history embeddings, uses a frozen LLM with a learnable head $f_\theta$ to predict $\hat{x}_t$; a prototype $p$ supplies a stability prior, and the anomaly score combines reconstruction error ($d_{L2}$) and prototype misalignment ($d_{cos}$) to trigger self-correction.

## 3 METHODOLOGY

As introduced in Section 2, we cast step-level error detection in LLM-based MAS as history-conditioned, *unsupervised* anomaly detection. Fig. 2 overviews MASC, which performs *real-time, unsupervised error detection and correction*. The central idea is to learn a compact model of *normal* multi-agent behavior and flag steps that deviate from this learned pattern.

For each step $t$, MASC executes three stages: (1) **Contextual Encoding**, which converts raw inputs (task query, agent roles, and interaction history) into task-aware vector embeddings; (2) **Prototype-Guided Reconstruction**, our detection module, which predicts the current step's embedding from historical context and identifies anomalies via reconstruction residuals and deviation from a learned prototype of normality; and (3) **Anomaly-Triggered Self-Correction**, wherein a high anomaly score triggers a dedicated *Correction Agent* to revise the flagged output and write back the corrected result to the shared history.

### 3.1 CONTEXTUAL ENCODING

Assume we have a set of agent roles $\{\mathcal{R}_i\}_{i=1}^N$ for $N$ agents. At each time step $t$, the input to our detector consists of the task query $\mathcal{Q}$ and the Agent role–output history $\mathcal{H}_{t-1}$ from previous steps. Here, $\mathcal{H}_{t-1}$ records, for each prior agent/tool call $j$, the pair comprising the acting agent's role and its emitted output: $\mathcal{H}_{t-1} = \{(\mathcal{R}_j, \mathcal{O}_j)\}_{j=1}^{t-1}$. We begin by encoding these symbolic components into dense vector representations using a pre-trained encoder, denoted as $\text{Embed}(\cdot)$. Formally, this tokenization process is defined as:

$$\mathbf{q} = \text{Embed}(\mathcal{Q}), \tag{2}$$

$$\mathbf{r}_i = \text{Embed}(\mathcal{R}_i), \quad i = 1, \dots, N, \tag{3}$$

$$\mathbf{h}_j = \mathbf{r}_j \| \text{Embed}(\mathcal{O}_j), \quad j = 1, 2, \dots, t-1 \tag{4}$$

Here, $\mathbf{q}$ is the embedding of the task query, $\mathbf{r}_i$ is the embedding of the $i$-th agent role description, and $\mathbf{h}_j$ is the embedding of the $i$-th historical conversation, obtained by concatenating the role and response embeddings of agent at step $j$. Subsequently, $\mathbf{q}$ and $\mathbf{h}_j$ are passed through separate, learnable linear projection layers to map them into a unified hidden dimension:

$$\tilde{\mathbf{q}} = f_q(\mathbf{q}), \quad \tilde{\mathbf{h}}_j = f_h(\mathbf{h}_j), \tag{5}$$

where $f_q, f_h$ are learnable linear projections. This contextual encoding step produces task-adapted representations $\tilde{\mathbf{q}}$ and $\tilde{\mathbf{h}}$ that fuse the necessary information for the downstream task.

### 3.2 PROTOTYPE-GUIDED RECONSTRUCTION

The core of our detection mechanism is the principle of reconstruction-based anomaly detection. The underlying intuition is that a model trained exclusively on normal data can reconstruct valid

samples with high fidelity, whereas its ability to reconstruct anomalous samples is inherently weaker. This discrepancy can be exploited to identify errors. However, directly transplanting such methods to step-level anomaly detection in MAS is non-trivial. Unlike image or time series domains where anomalies often exhibit strong signal deviation, abnormal steps in MAS are often semantically close to normal steps and only become erroneous under specific execution contexts. This weak semantic separability makes context-aware modeling crucial.

**Next-Execution Reconstruction.** To address this, we propose a **Next-Execution Reconstruction** module. Instead of reconstructing the input, we leverage the causal structure of agent interactions. Given the history up to step $t-1$, the module predicts the representation of the *next execution step*, $t$. This forces the model to learn the causal dependencies that govern normal interaction flow. We employ a pre-trained, frozen Large Language Model (LLM) to encode the context sequence. Its output is then passed through a learnable linear projection layer, denoted $f_\theta$, to generate the final prediction. Formally, the prediction, $\hat{\mathbf{x}}_t$, is generated by feeding the projected query and history embeddings into our model:

$$\hat{\mathbf{x}}_t = f_\theta\Big( \text{LLM} \big( \tilde{\mathbf{q}}, \tilde{\mathbf{h}}_1, \ldots, \tilde{\mathbf{h}}_{t-1} \big) \Big). \tag{6}$$

The projection layer $f_\theta$ maps the LLM's hidden representation back to the dimension of the raw history embedding $\mathbf{h}_j$. Anomalous steps, by violating causal consistency, will naturally exhibit a higher deviation between the prediction $\hat{\mathbf{x}}_t$ and the realized embedding, which we define as the ground truth $\mathbf{x}_t := \mathbf{h}_t$. For the first step ($t = 0$), the input sequence to the LLM consists solely of the projected query, $\tilde{\mathbf{q}}$.

**Prototype-Guided Enhancement.** While next-execution reconstruction is effective, it can be less reliable in early steps where the historical context is sparse. To mitigate this, we introduce a *prototype-guided enhancement* mechanism. We maintain a learnable prototype vector $\mathbf{p} \in \mathbb{R}^d$ that represents the centroid of normal step embeddings and acts as a stable anchor of normality. $d$ is the shared dimension of $x_t$, $\hat{x}_t$, and $h_j$. Given a normal trajectory, we collect the reconstructed embeddings $\hat{\mathbf{X}} = [\hat{\mathbf{x}}_1, \ldots, \hat{\mathbf{x}}_T]^\top \in \mathbb{R}^{T \times d}$ from our predictor and refine $\mathbf{p}$ via a single-head attention update that uses $\mathbf{p}$ as the query and $\hat{\mathbf{X}}$ as keys/values:

$$\mathbf{p} \leftarrow \text{Attn}\big(\mathbf{p}W_q, \hat{\mathbf{X}}W_k, \hat{\mathbf{X}}W_v\big) = \text{Softmax}\left( \frac{(\mathbf{p}W_q)(\hat{\mathbf{X}}W_k)^\top}{\sqrt{d}} \right) (\hat{\mathbf{X}}W_v), \tag{7}$$

where $W_q, W_k, W_v \in \mathbb{R}^{d \times d}$ are learnable projections. The prototype $\mathbf{p}$ is learnable and can be initialized from a Gaussian distribution or from the mean of pseudo-normal embeddings obtained by prompting the LLM. This design encourages each reconstructed step to align with the prototype center, providing robustness when contextual history is scarce or noisy.

## 3.3 TRAINING OBJECTIVE

Our framework is trained in a *fully unsupervised manner* using only normal trajectories, avoiding costly step-level error annotations. The objective combines a reconstruction loss that enforces causal consistency with a prototype loss that regularizes reconstructed steps toward the center of the normal distribution.

**Reconstruction Loss.** For a trajectory of length $T$, our predictor produces the *current-step* representation $\hat{\mathbf{x}}_t$ conditioned on the context up to step $t-1$. We minimize the mean squared error between the predicted and realized embeddings on normal data:

$$\mathcal{L}_{\text{recon}} = \frac{1}{T} \sum_{t=1}^{T} \big\| \hat{\mathbf{x}}_t - \mathbf{x}_t \big\|_2^2. \tag{8}$$

**Prototype Loss.** To enhance robustness when history is short or noisy, we introduce a learnable prototype vector $\mathbf{p}$ that encodes the central tendency of normal steps. We regularize each reconstructed embedding toward $\mathbf{p}$ via cosine similarity:

$$\mathcal{L}_{\text{proto}} = \frac{1}{T} \sum_{t=1}^{T} \Big( 1 - \cos\big(\hat{\mathbf{x}}_t, \mathbf{p}\big) \Big). \tag{9}$$

**Total Loss.** The final training objective is a weighted combination of the two terms:

$$\mathcal{L} = \mathcal{L}_{\text{recon}} + \lambda \mathcal{L}_{\text{proto}}, \tag{10}$$

where $\lambda$ balances the reconstruction fidelity and prototype alignment. Since both terms are defined solely on normal trajectories, the framework naturally learns to distinguish abnormal steps at inference time as those that yield larger residuals or weaker alignment with the prototype.

### 3.4 INFERENCE AND ANOMALY SCORING

At test time, we assign an score to the *current* step $t$ immediately after its output is produced. Given the context up to step $t-1$, our predictor generates the current-step reconstruction $\hat{\mathbf{x}}_t$. We then combine an L2 reconstruction error with a cosine-based prototype misalignment:

$$s(t) = \alpha \left\| \hat{\mathbf{x}}_t - \mathbf{x}_t \right\|_2^2 + \beta \left( 1 - \cos(\hat{\mathbf{x}}_t, \mathbf{p}) \right) \tag{11}$$

where $\alpha$ and $\beta$ are weighting hyperparameters. A higher score indicates a greater likelihood of anomaly. This design preserves the strength of LLM-based reconstruction in capturing causal consistency across steps, while leveraging the prototype as a stable reference, which is especially helpful when contextual history is scarce.

### 3.5 SELF-CORRECTION VIA ANOMALY-TRIGGERED INTERVENTION

The final stage of our framework is a self-correction mechanism initiated by our anomaly detector. At each step $t$, if the output $\mathcal{O}_t$ from agent $i = \phi(t)$ yields an anomaly score $s(t)$ exceeding a threshold $\delta$, an intervention is triggered. This gating mechanism ensures corrections are targeted and efficient, preventing error propagation. In contrast to re-invoking the original agent, we employ a *dedicated correction agent* with policy $\pi_{\text{corr}}$. When triggered, correction agent is prompted with the current context and a correction instruction to revise the flagged output. The final, potentially corrected, output $\tilde{\mathcal{O}}_t$ is determined by:

$$\tilde{\mathcal{O}}_t = \begin{cases} \tilde{\mathcal{O}}_t, & \text{if } s(t) \leq \delta, \\ \pi_{\text{corr}}(\mathcal{H}_{t-1}, \mathcal{O}_t, \mathcal{P}_{\text{corr}}), & \text{if } s(t) > \delta, \end{cases} \tag{12}$$

where $\mathcal{H}_{t-1}$ is the textual conversation history up to step $t - 1$ and $P_{\text{corr}}$ is a correction instruction that requests reconsideration. The corrected output $\tilde{\mathcal{O}}_t$ replaces the original, thereby updating the history that subsequent agents receive (i.e., $\mathcal{H}_t$ will contain $\tilde{\mathcal{O}}_t$). This self-healing loop mitigates errors at their source and prevents cascading errors.

## 4 EXPERIMENTS

We evaluate our proposed framework from two perspectives: (1) the effectiveness of our unsupervised anomaly detector for step-level error detection, and (2) the end-to-end performance improvement when integrating our MASC framework into existing multi-agent systems. This section is organized as follows: We first detail the experimental setup for both tasks. Next, we present the main results for step error detection and framework integration. Finally, we provide in-depth ablation studies to analyze the contribution and effectiveness of our framework. Experiments on hyperparameters and prototype updates are provided in the Appendix C

### 4.1 EXPERIMENTAL SETUP

**Datasets and Tasks.** For the error detection task, we use the `Who&When` benchmark (Zhang et al., 2025c), which includes a *handcrafted* and an *automated* subset. We evaluate under two conditions: **w/ GT** (with access to the ground-truth answer of the query) and **w/o GT** (relying only on agent logs). To assess the end-to-end performance of our integrated framework, we evaluate it on six standard benchmarks spanning three domains: general reasoning (`MMLU` (Hendrycks et al., 2021)), mathematical problem solving (`GSM8K` (Cobbe et al., 2021), `AQuA` (Ling et al., 2017), `MultiArith` (Roy & Roth, 2016), and `SVAMP` (Patel et al., 2021)), and code generation(`HumanEval` (Chen et al., 2021)). For all evaluations, we use the official data splits. Detailed statistics provided in the Appendix A.

Table 1: Performance (%) comparison on `Who&When` benchmark.

| Model | Who&When (handcraft) | | Who&When (automated) | |
|---|---|---|---|---|
| | w/ GT | w/o GT | w/ GT | w/o GT |
| All-at-Once | 44.98/6.90 | 47.15/10.34 | 30.26/14.29 | 32.16/9.52 |
| Step-by-Step | 58.25/15.87 | 50.74/14.29 | 39.66/13.79 | 30.42/8.62 |
| Binary-Search | 54.81/13.49 | 51.93/15.52 | 24.39/5.17 | 21.97/6.90 |
| Bert Classifier | 60.58/10.37 | 72.86/13.79 | 62.91/15.21 | 67.15/13.68 |
| LLM Classifier | 63.12/17.93 | 65.79/18.96 | 65.50/17.34 | 65.39/16.95 |
| **MASC** | **69.10/18.25** | **77.84/20.79** | **69.62/18.79** | **75.62/21.72** |

**Baselines.** For step-level error detection, we consider two representative categories of baselines: (1) *LLM-as-detector*, directly prompts large language models to judge whether a step is erroneous, following the strategies provided in the `Who&When` benchmark (Zhang et al., 2025c), including All-at-Once, Step-by-Step, and Binary Search. (2) *strong supervised models*, including a sentence classification model based on `BERT` (Koroteev, 2021) and another classifier that uses a large language model encoder. For the latter, we represent each sentence by taking the hidden state of its final token and pass it to a trainable MLP classifier head (BehnamGhader et al., 2024).

Beyond detection, we also evaluate the effect of integrating our MASC into MAS frameworks. We consider a broad range of baselines covering both single-agent prompting strategies and multi-agent communication: 1) *single-agent methods* namely Chain-of-Thought (CoT) (Wei et al., 2022b) and Self-Consistency (SC) (Wang et al., 2022); 2) *multi-agent systems with fixed topologies* including Chain, Complete Graph, Random Graph (Qian et al., 2024), and LLM-Debate (Du et al., 2023).

**Implementation.** All methods are evaluated on the same data split: $20\%$ of the trajectories are used for training (when applicable) and the remaining $80\%$ are reserved for testing. For step error detection, we compare three categories of baselines: (1) *LLM-as-detector* methods using `GPT-4o-mini`, following the official `Who&When` benchmark (Zhang et al., 2025c) for prompts and evaluation protocols; (2) *supervised models*, where a sentence bert (`all-MiniLM-L6-v2`)and open-source LLM (`LLaMA-3.1-8B-Instruct`) are used as frozen encoders with a trainable MLP classifier head; and (3) our method, in which queries, role descriptions, and historical responses are encoded using `all-MiniLM-L6-v2`, and `LLaMA-3.1-8B-Instruct` serves as the frozen backbone LLM for next-execution reconstruction, with only the projection layers and prototype module being trainable. For framework integration, all agents are instantiated with `GPT-4o-mini`, and the overall implementation strictly follows the settings of G-Designer (Zhang et al., 2024a). Additional hyperparameters and training details are provided in the Appendix B.

**Metrics.** For error detection, we report AUC-ROC and step-level accuracy for localization precision. For framework integration, we report the final task accuracy (%) on each benchmark.

## 4.2 MAIN RESULTS

**Step-Level Error Detection.** Table 1 shows that our unsupervised detector significantly outperforms all baselines, including supervised ones. In the challenging 'w/o GT' setting, our method achieves an AUC-ROC of **77.84%** on the handcrafted data and **75.62%** on the automated data. These results demonstrate the superiority of our reconstruction-based approach in modeling the dynamics of agent interactions without needing any error labels.

**MASC Integration with Existing Frameworks.** We further evaluate the practical impact of our full framework by integrating it into existing MAS. As shown in Table 2, MASC consistently enhances performance across all tested frameworks. On average, it yields a **1.29%** performance gain. For instance, when applied to the powerful LLM-Debate framework, it improves the average accuracy from 87.53% to 88.89%. This confirms that our real-time detection and correction mechanism is effective at mitigating cascading errors and improving overall system robustness.

## 4.3 ABLATION STUDIES

Table 2: Performance comparison (%) on six benchmarks. Our framework, MASC, consistently improves performance when integrated with various MAS architectures. 'M.Arith' and 'H.Eval' are abbreviations for `MultiArith` and `HumanEval`, respectively.

| Method | MMLU | GSM8K | AQuA | M.Arith | SVAMP | H.Eval | Avg. |
|---|---|---|---|---|---|---|---|
| Vanilla | 80.39 | 82.30 | 71.06 | 93.09 | 86.55 | 71.39 | 80.80 |
| CoT | 81.69 | 86.50 | 73.58 | 93.25 | 87.36 | 74.67 | 82.84 |
| SC (CoT) | 83.66 | 81.60 | 75.63 | 94.12 | 88.59 | 79.83 | 83.91 |
| Chain | 83.01 | 88.30 | 74.05 | 93.27 | 87.17 | 81.37 | 84.53 |
| Complete | 82.35 | 86.10 | 72.95 | 94.53 | 84.01 | 79.03 | 82.16 |
| Random | 84.31 | 86.90 | 76.48 | 94.08 | 87.54 | 82.66 | 85.33 |
| Debate | 84.96 | 91.40 | 77.65 | 96.36 | 90.11 | 84.70 | 87.53 |
| MASC (Chain) | 83.57 ↑0.56 | 90.51 ↑2.21 | 76.23 ↑2.18 | 93.96 ↑0.69 | 88.54 ↑1.37 | 82.91 ↑1.54 | **85.95** |
| MASC (Complete) | 84.07 ↑1.72 | 89.25 ↑3.15 | 74.11 ↑1.16 | 95.04 ↑0.51 | 85.26 ↑1.25 | 81.37 ↑2.34 | **84.85** |
| MASC (Random) | 85.29 ↑0.98 | 88.91 ↑2.01 | 77.12 ↑0.64 | 94.82 ↑0.74 | 88.29 ↑0.75 | 84.01 ↑1.35 | **86.41** |
| MASC (Debate) | 86.11 ↑1.15 | 93.39 ↑1.99 | 79.21 ↑1.56 | 97.15 ↑0.79 | 91.26 ↑1.15 | 86.23 ↑1.53 | **88.89** |

**Analysis of the Detection Module.** We analyze the contribution of the two core components of our detector: next-execution reconstruction and prototype guidance. Fig. 3 shows the results. Removing the reconstruction objective causes a substantial drop in accuracy, as the model loses the ability to capture causal dependencies between steps. Similarly, removing the prototype mechanism harms performance, especially in early steps where historical context is limited, confirming its role in providing a stable reference for normality. Both components are thus essential for reliable detection.

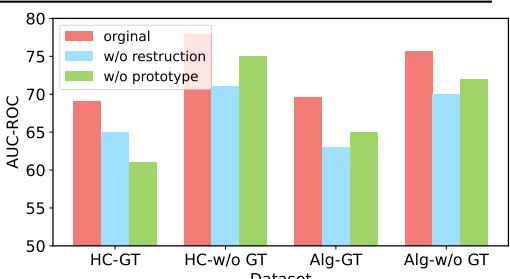

Figure 3: Ablation of reconstruction and prototype modules on `Who&When`.

**Impact of Detection on Downstream Correction.** Building on Table 1, where detector quality ranks *Step-by-Step ≺ BERT classifier ≺ LLM classifier ≺ MASC*, we evaluate how this ordering translates to correction gains on GSM8K under three MAS topologies (Chain, Complete, Random). Table 3 shows that the ranking largely carries over to end-to-end correction. *Step-by-Step* hurts average performance ($-0.86$), and the *LLM classifier* fails to transfer its advantage, even degrading in denser settings ($-0.85$). The *BERT classifier* yields small but unstable gains ($+0.49$). In contrast, MASC consistently improves across all topologies, with up to $+3.15$ and an average of $+2.46$ over vanilla (no detection/correction), underscoring the importance of robust detection for effective downstream correction.

Table 3: Performance of MASC and other error detection methods on downstream correction (GSM8K).

| Method | Chain | Complete | Random | Average |
|---|---|---|---|---|
| Vanilla | 88.30 | 86.10 | 86.90 | 87.10 |
| MASC | 90.51 ↑2.21 | 89.25 ↑3.15 | 88.91 ↑2.01 | 89.56 ↑2.46 |
| Step-by-Step | 87.23 ↓1.07 | 84.29 ↓1.81 | 87.21 ↑0.31 | 86.24 ↓0.86 |
| BERT Classifier | 89.12 ↑0.82 | 85.12 ↓0.98 | 88.53 ↑1.63 | 87.59 ↑0.49 |
| LLM Classifier | 87.65 ↓0.65 | 83.27 ↓2.83 | 87.82 ↑0.92 | 86.25 ↓0.85 |

### 4.4 Score Distribution Analysis

An effective anomaly detector should assign clearly distinguishable scores to normal and erroneous steps, such that a simple threshold can separate the two distributions. To examine whether our method achieves this property, we visualize the score distributions of normal versus error steps on the `Who&When` *(automated, w/o GT)* setting. As shown in Fig. 4, the baseline method that directly applies BERT embeddings produces highly overlapping distributions, making it difficult to discriminate between correct and erroneous steps. In contrast, our proposed approach yields a much larger separation: normal steps concentrate on higher confidence scores, while error

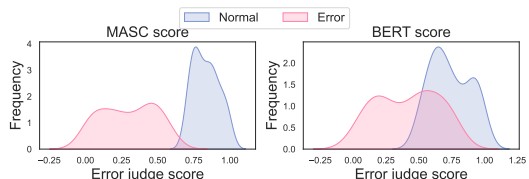

Figure 4: Normal vs. error score distributions on `Who&When`: MASC (left) vs. BERT (right); MASC shows a cleaner separation.

steps shift toward lower values. This clear gap confirms that our reconstruction–prototype framework captures the causal structure of multi-agent reasoning, enabling robust error detection with a simple thresholding mechanism. The complete results and additional dataset visualizations are provided in the appendix.

# 5 RELATED WORK

## 5.1 LLM-BASED MULTI-AGENT SYSTEMS

Recent advances in large language model (LLM)-based multi-agent systems (MAS) have demonstrated strong capabilities across diverse reasoning and decision-making tasks (He et al., 2025a; Zhang et al., 2024b; Yi et al., 2024; Ishibashi & Nishimura, 2024). The effectiveness of these systems stems from collaboration among heterogeneous agents, where role specialization and structured communication strategies can significantly enhance overall performance (Li et al., 2023; Xie et al., 2024; Shen et al., 2025; Li et al., 2025). Early implementations of LLM-based MAS were largely *handcrafted*, where system designers manually specified agent roles, prompts, and communication topologies (Wu et al., 2023b; Li et al., 2023; Qian et al., 2023). Such systems demonstrated the potential of LLM-based collaboration but required extensive manual design effort, limiting scalability and adaptability (Zhang et al., 2025b). To overcome these limitations, more recent research has explored *automated* approaches. Examples include frameworks that automate agent role assignment (Dang et al., 2025; Chen et al., 2025) or adaptively construct inter-agent topologies (Zhang et al., 2024a), thereby reducing reliance on fixed human-designed rules. The most recent line of work moves toward *fully automated* MAS, in which both role specialization and communication structures evolve dynamically during execution (Nie et al., 2025; Zhang et al., 2025a;d). However, as automation increases, so too does the risk of uncontrolled error propagation and vulnerability to adversarial perturbations, highlighting the need for robustness-oriented research.

## 5.2 ROBUST MULTI-AGENT SYSTEMS

Despite their promise, LLM-based MAS face significant robustness challenges. Recent studies have highlighted that failures in MAS often stem from error propagation across agents, adversarial prompt injections, and compromised communication protocols (Zhan et al., 2024; Chen et al., 2024; Andriushchenko et al., 2025). These vulnerabilities can amplify individual agent errors into systemic failures, threatening the reliability of downstream decision-making (Gan et al., 2024; Yuan et al., 2024). Research on security has identified message-passing mechanisms as a critical attack surface (Yu et al., 2024), while trust frameworks such as A-Trust (He et al., 2025b) and G-Safeguard (Wang et al., 2025) focus on detecting compromised agents through network analysis or trust dimension modeling. Parallel to this, the failure attribution literature seeks to explain why and where MAS fail. For instance, MAST (Cemri et al., 2025) provided a taxonomy of fourteen error patterns, and the Who&When benchmark (Zhang et al., 2025c) systematically annotated erroneous steps within multi-agent trajectories to enable step-level failure analysis. These efforts underscore that achieving robustness in MAS requires not only stronger anomaly detection but also mechanisms for self-correction and resilience against cascading errors.

# 6 CONCLUSION

In this work, we introduce MASC, a metacognitive layer for LLM-based multi-agent systems that performs *real-time, unsupervised, step-level error detection and targeted self-correction*. By reframing detection as history-conditioned anomaly scoring with Next-Execution Reconstruction and a Prototype-Guided stability prior, MASC reliably identifies deviations even in context-scarce early steps and intervenes before errors cascade. Empirically, MASC attains substantial AUC-ROC gains on the step-level error detection of MAS and delivers consistent end-to-end improvements when plugged into diverse MAS architectures across six standard benchmarks, demonstrating robustness with minimal overhead and broad plug-and-play utility. Notably, MASC is label-free and architecture-agnostic, enabling drop-in integration without retraining task policies. We hope this metacognitive layer serves as a reliability primitive for scalable, trustworthy multi-agent LLM systems.

ETHICS STATEMENT

This work is strictly limited to scientific investigation and does not involve human subjects, animals, or environmentally sensitive materials. Consequently, it raises no ethical concerns or conflicts of interest. Throughout the study, we adhere to established principles of scientific integrity and ethical research practices to ensure the rigor, reliability, and validity of our findings.

REPRODUCIBILITY STATEMENT

All components of MASC are designed to ensure reproducibility. The overall task and evaluation setup are formally defined in the main text. Dataset statistics and annotation protocols are reported in Appendix A, implementation details are provided in Appendix B, hyperparameter analyses are included in Appendix C, and full pseudocode is presented in Appendix D. All experiments are conducted under standardized evaluation criteria, and baseline results have been carefully verified to guarantee fairness and consistency. The code are provide in https://anonymous.4open.science/r/MASC-6A03/.

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

## A DATASET STATISTIC

We present the dataset statistics in Table 4, which is as same experimental setup as G-Designer Zhang et al. (2024a).

Table 4: Dataset descriptions and statistics.

| Category | Dataset | Answer Type | Metric | #Test | License |
|---|---|---|---|---|---|
| General reasoning | MMLU | Multi-choice | Acc. | 153 | MIT License |
| Math reasoning | GSM8K | Number | Acc. | 1,319 | MIT License |
| | MultiArith | Number | Acc. | 600 | Unspecified |
| | SVAMP | Number | Acc. | 1,000 | MIT License |
| | AQuA | Multi-choice | Acc. | 254 | Apache-2.0 |
| Code generation | HumanEval | Code | Pass@1 | 164 | MIT License |

## B IMPLEMENTATION DETAILS

**Training Details.** For the *LLM-as-detector* baselines, we directly adopt the official implementation from the `Who&When` benchmark (Zhang et al., 2025c), including both code and prompts, and evaluate the All-at-Once, Step-by-Step, and Binary Search variants without modification. For *strong supervised models* and our proposed MASC, which require training, we use Adam as the optimizer and perform random search over training-related hyperparameters to ensure fair comparison; the final values are reported in Table 5. Both supervised baselines are trained on individual steps, by mixing all steps from the traces provided in `Who&When` and shuffling them into mini-batches. In contrast, MASCoperates over full trajectories, leveraging historical context to perform autoregressive reconstruction. All experiments are run under a consistent setup to ensure reproducibility.

Table 5: Hyperparameter settings for different methods across Who&When datasets.

| Method | Hyperparameter | HC w/ GT | HC w/o GT | Auto w/ GT | Auto w/o GT |
|---|---|---|---|---|---|
| **BERT Classifier** | epochs | 50 | 50 | 50 | 50 |
| | lr | 1e-5 | 1e-5 | 2e-5 | 2e-5 |
| | weight decay | 0.01 | 0.01 | 0.01 | 0.01 |
| | batch Size | 32 | 32 | 64 | 64 |
| | hidden Size | 384 | 384 | 384 | 384 |
| **LLaMA Classifier** | epochs | 8 | 10 | 6 | 8 |
| | lr | 5e-5 | 5e-5 | 1e-4 | 1e-4 |
| | weight decay | 0.05 | 0.05 | 0.05 | 0.05 |
| | batch Size | 50 | 50 | 50 | 50 |
| | hidden Size | 4096 | 4096 | 4096 | 4096 |
| **MASC** | epochs | 10 | 10 | 5 | 5 |
| | lr | 1e-4 | 1e-4 | 5e-5 | 5e-5 |
| | weight decay | 0 | 0 | 0 | 0 |
| | batch Size | - | - | - | - |
| | hidden Size | 384 | 384 | 384 | 384 |

**Recovery Prompt for Error Correction**    This prompt is designed to support error recovery in our multi-agent reasoning framework. When a step is flagged by the anomaly detector as potentially incorrect, the responsible agent is asked to re-examine its previous response in light of the original query and the available context. The prompt enforces strict reflection rules, requiring the agent either to confirm the correctness of its earlier output or to provide a corrected version, and mandates a fixed JSON format for consistency. This ensures that correction is explicit, structured, and directly usable for downstream evaluation and analysis.

---

PROMPT FOR RESPONSE RECOVERY

You are an AI agent playing the role of "`{agent.role}`". You previously generated a response during a multi-agent reasoning process, but an anomaly detector flagged your output as potentially incorrect. Your task is to carefully reflect on whether your earlier response was indeed wrong given the original query and the current context.

Please follow these rules strictly:

1. Re-examine the original query and your earlier response in the context of your role.

2. If after reflection you believe your previous response is correct and does not require modification, explicitly state that no correction is needed.

3. If you identify errors or find a better answer, provide a corrected response.

4. Always output in the fixed JSON format below. Do not add extra explanations outside the JSON.

**Output format:**

```
{
  "correction_needed": "Yes" or "No",
  "final_response": "If correction_needed=No,

  repeat your original response here.

  If Yes, provide the corrected response."
}
```

**Input Information:**

- Query: {question}
- Your Previous Response: {mas.history}
- Context (previous steps if available): {agent.spatialinfo()}

---

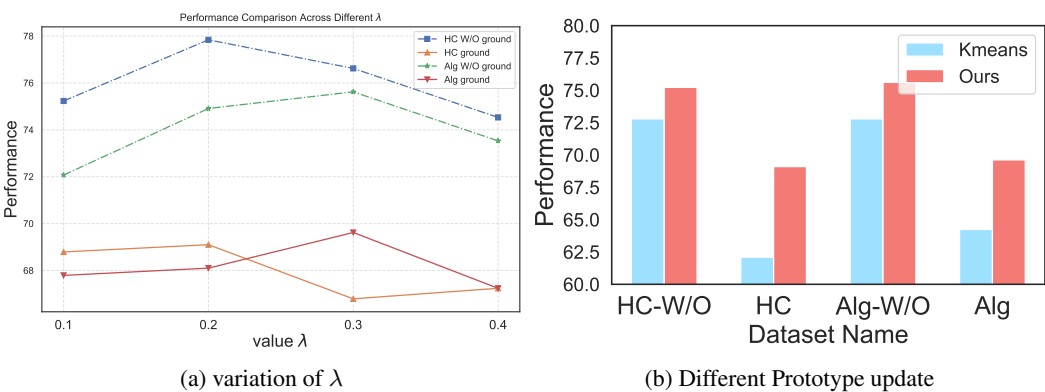

(a) variation of $\lambda$          (b) Different Prototype update

Figure 5: Hyperparameter and prototype updating analysis.

## C  ANALYSIS OF HYPER-PARAMETERS

**Hyperparameter Sensitivity of $\lambda$**  As shown in Fig. 5a, our method is generally insensitive to the choice of $\lambda$, achieving stable performance across a wide range of values. Notably, the optimal $\lambda$ differs between datasets: Hand-Crafted trajectories perform best near $\lambda = 0.2$, while Algorithm-Generated data favors larger values (e.g., $\lambda = 0.3$), likely because errors in the latter tend to occur earlier, making the prototype component more critical when historical context is limited. Overall, these results indicate that while tuning $\lambda$ can yield slight gains, our framework remains robust without heavy dependence on this hyperparameter.

**Prototype Updating.**  As shown in Fig. 5b, our attention-based prototype updating mechanism consistently surpasses the KMeans clustering baseline across all settings. The limitation of KMeans lies in its reliance on static, distance-based centroids, which cannot adequately capture contextual dependencies or the dynamic nature of multi-agent interactions. In contrast, our method leverages an attention mechanism to adaptively refine the prototype vector at each step, ensuring that it remains aligned with the evolving distribution of normal trajectories. This adaptive updating leads to more reliable discrimination, yielding superior performance both with and without ground-truth supervision.

## D  PSEUDOCODE

The training algorithm of MASCis shown in Algorithm 1. After training, the resulting detector is integrated into the MAS execution process, where it continuously monitors agent outputs and triggers the correction agent when anomalies are detected, thus enabling real-time self-correction during collaboration. The pseudo-code for this process is shown in Algorithm. 2.

## E  THE USE OF LARGE LANGUAGE MODELS

To enhance readability, we employed OpenAI GPT-5 strictly as a language editing tool for grammar correction and stylistic refinement. Its use was limited to functions analogous to conventional proofreading and did not contribute to the conception, methodology, analysis, or scientific content of this work.

---

**Algorithm 1:** Unsupervised Training of MASC

---

**Input:** Normal trajectories $\{\mathcal{H}_i\}_{i=1}^M$, hyper-parameter $\lambda$, A LLM with frozen parameters
**Output:** Trained parameters of $f_q$, $f_h$, $f_\theta$, $\mathrm{Attn}$ and prototype $\mathbf{p}$
**for** *trajectory $\mathcal{H}_i \in \{\mathcal{H}_i\}_{i=1}^M$* **do**
     Initialize $\mathcal{L}_{\mathrm{recon}} = 0$, $\mathcal{L}_{\mathrm{proto}} = 0$, $T = \mathbf{length}(\mathcal{H}_i)$ ;
     **for** $t = 1$ **to** $T$ **do**
         Encode query $\mathcal{Q}$, the role of current agent $\mathcal{R}$ and history $\mathcal{H}_{t-1}$ into $\tilde{\mathbf{q}}$, $\tilde{\mathbf{h}}$ via Eq. 2 and 5;
         Predict $\hat{\mathbf{x}}_t$ via Eq. 6;
         Get ground truth $\mathbf{x}_t = \mathbf{h}^{(t)}$;
         // Update losses
         $\mathcal{L}_{\mathrm{recon}} \leftarrow \mathcal{L}_{\mathrm{recon}} + \|\hat{\mathbf{x}}_t - \mathbf{x}_t\|_2^2$;
         Calculate prototype $\mathbf{p}$ via Eq 7
         $\mathcal{L}_{\mathrm{proto}} \leftarrow \mathcal{L}_{\mathrm{proto}} + (1 - \cos(\hat{\mathbf{x}}_t, \mathbf{p}))$;
     // Final loss
     $\mathcal{L} = \frac{1}{T}\mathcal{L}_{\mathrm{recon}} + \lambda \cdot \frac{1}{T}\mathcal{L}_{\mathrm{proto}}$;
     Update all learnable parameters and prototype $\mathbf{p}$ by $\nabla_\theta \mathcal{L}$;

---

**Algorithm 2:** Real-Time Self-Correction via Anomaly-Triggered Intervention

---

**Input:** A LLM-based Multi-agent System $\mathcal{M}$ with $N$ nodes, hyper-parameter $\lambda$, A LLM with
         frozen parameters, Query $\mathcal{Q}$
**Output:** Normal trajectory $\mathcal{H} = \{h_0 \dots h_t\}$ after self correction (if necessary)
**for** *node $t \in \{1, 2, \dots, N\}$* **do**
     Encode query $\mathcal{Q}$, the role of current agent $\mathcal{R}$ and history $\mathcal{H}_{t-1}$ into $\tilde{\mathbf{q}}$, $\tilde{\mathbf{h}}$ via Eq. 2 and 5;
     Predict $\hat{\mathbf{x}}_t$ via Eq. 6;
     Get ground truth $\mathbf{x}_t = \mathbf{h}_t$;
     Calculate Anomaly Score $s(t)$ with $\hat{\mathbf{x}}_t$, $\mathbf{x}_t$, $\mathbf{p}$ via Eq. 11
     Update the current output $\mathcal{O}_t$ via Eq. 12 to $\tilde{\mathcal{O}}_t$ and add into the Normal trajectory $\mathcal{H}$

---

