# OpenReview forum: "Metacognitive Self-Correction for Multi-Agent System via Prototype-Guided Next-Execution Reconstruction"
_ICLR.cc/2026/Conference — ICLR 2026 Conference Withdrawn Submission_

### Official Review · Reviewer_QXe9 · 2025-10-27

**Soundness:** 2
**Presentation:** 3
**Contribution:** 3
**Rating:** 2
**Confidence:** 4

**Summary:**

This paper proposes a metacognitive framework for detecting and correcting errors in LLM-based multi-agent systems using unsupervised anomaly detection. The method combines next-execution reconstruction and an anomaly detection for detecting potential error steps. While the application domain is relevant, the technical contributions are largely incremental, applying established anomaly detection techniques to multi-agent trajectories with modest empirical improvements.

**Strengths:**

1. Addresses a real problem in multi-agent systems - error propagation can significantly degrade performance

2. Unsupervised approach avoids the need for expensive step-level error annotations

3. Experiments setup is diverse but ablation studies can be improved.

**Weaknesses:**

1. While the error propagation is understandable, in MAS it uses responses from prompt-based attack to demonstrate cascading failures don't necessarily reflect realistic error modes.

2. "Next-Execution Reconstruction" is basically predicting the next item in a sequence while the prototype component is similar to existing work on prototype networks and center loss

3. Limited technical novelty overall, the core approach is standard reconstruction-based anomaly detection [1] with minor adaptations ( prototype loss ).

4. The paper might have a reproducibility issue where it does not report values for α, β, or δ despite claiming hyperparameters are in Appendix B. The 3 parameters are crucial in determining if the step is anomaly or not.

[1]: Hou, Jinlei, et al. "Divide-and-assemble: Learning block-wise memory for unsupervised anomaly detection." Proceedings of the IEEE/CVF International Conference on Computer Vision. 2021.

**Questions:**

Major:

1. How does the method compare to simpler alternatives like confidence-based ( from logits ) filtering or majority voting among agents?

2. Can you provide analysis of actual error types detected versus false positives? What kinds of errors does the method miss?

3. What are the anomaly thresholds δ, alpha and beta used in all experiments ( also how are the selected ) ?

4. Any potential computational overhead to find this threshold?

Minor:

5. Is there any representation feature learned via reconstruction loss and prototype loss yielded strong representations which can distinguish between in distribution and anomaly step?

6. Would using a stronger LLM as a detector yield better results?

7. Why is the batch size not provided in Appendix B, Table 5?

8. What was the logic behind the bold average for the MASC score? MASC (Complete) only yielded an average of 84.85 which performs worse than Random and Debate methods.

---

### Official Review · Reviewer_kLWu · 2025-10-30

**Soundness:** 3
**Presentation:** 3
**Contribution:** 3
**Rating:** 4
**Confidence:** 3

**Summary:**

This paper introduces MASC, a metacognitive framework for real-time, unsupervised error detection and self-correction in Large Language Model (LLM)-based Multi-Agent Systems (MAS). The core problem addressed is the brittleness of MAS to cascading errors, where a single faulty step can derail the entire problem-solving trajectory. MASC's main contributions are two complementary mechanisms: Next-Execution Reconstruction and Prototype-Guided Enhancement. When an anomaly is detected, MASC triggers a dedicated correction agent to revise the output before it propagates. The entire framework is trained in an unsupervised manner on normal trajectories.

**Strengths:**

- The paper tackles a critical and timely problem. As LLM-based multi-agent systems become more complex, their susceptibility to cascading errors becomes a major bottleneck for reliable deployment. Developing robust, real-time error detection and correction mechanisms is of high importance to the community.

- The core idea of Next-Execution Reconstruction is novel and insightful. Framing the problem as predicting the subsequent step's embedding, rather than auto-encoding the current one, is a clever way to explicitly model the causal flow and dependencies in agent interactions. The addition of the Prototype-Guided Enhancement is a well-motivated solution to the cold-start problem (sparse context in early steps), making the detector more robust.

- The experimental validation is comprehensive. The authors evaluate MASC on both a specialized step-level error detection benchmark (Who&When) and as a plug-in component in diverse MAS frameworks across six standard reasoning benchmarks. The outperformance against both prompting-based and supervised baselines is convincing. The ablation studies and score distribution analysis effectively demonstrate the contribution of each component and the detector's efficacy.

**Weaknesses:**

- The prototype mechanism models the distribution of all normal steps with a single learnable vector p. This assumes that normal agent actions are unimodally distributed around a single centroid. However, in complex, multi-stage tasks, valid actions can be highly diverse. For instance, a code generation step is semantically very different from a planning step or a verification step, yet all can be part of a single normal trajectory. A single prototype may not be expressive enough to capture this multi-modal nature, potentially leading to false positives or negatives for valid but less-common action types.

- The paper's contribution is positioned as both detection and self-correction. While the detection mechanism is thoroughly analyzed and ablated, the correction part is less explored. It relies on a dedicated correction agent prompted with a fixed template. This approach raises several questions: How was this agent/prompt designed? How does its performance compare to simpler baselines, such as simply re-prompting the original agent with a request to reconsider? The effectiveness of the entire MASC loop heavily depends on this correction step, yet its reliability and contribution are not independently analyzed. An error by the correction agent could be even more damaging.

- The abstract claims the framework mitigates errors with minimal overhead. However, the proposed method requires an additional forward pass through an LLM (the reconstruction backbone) and several smaller models at every single step of the MAS execution. For long and complex trajectories, this could introduce significant computational and latency overhead, which might challenge its suitability for real-time applications. The paper provides no quantitative analysis of this overhead compared to the vanilla MAS execution.

- The unsupervised training relies on a corpus of normal trajectories. The paper doesn't discuss how these trajectories are sourced or the potential impact of noise in this data. In practice, even successful trajectories might contain sub-optimal, inefficient, or slightly flawed intermediate steps. It is unclear how sensitive MASC's training is to such soft errors within the supposedly normal training data.

**Questions:**

- Could you comment on the limitation of using a single prototype vector p to represent the diverse space of normal agent actions? Have you considered alternatives like using multiple prototypes or a more expressive representation to better capture the potential multi-modality of normal steps?

- Could you provide more details and analysis on the self-correction component? Specifically, could you run an ablation study comparing the performance of the dedicated correction agent against a simpler baseline, such as re-prompting the original agent that made the error? This would help isolate the benefit of having a specialized corrector.

- The claim of minimal overhead is a key practical consideration. Could you please quantify the actual overhead introduced by MASC?

---

### Official Review · Reviewer_Gcnd · 2025-11-01

**Soundness:** 2
**Presentation:** 3
**Contribution:** 2
**Rating:** 2
**Confidence:** 3

**Summary:**

This paper presents MASC, a metacognitive framework designed for LLM-based Multi-Agent Systems. MASC enables real-time, unsupervised, step-level error detection and self-correction through two complementary mechanisms:
1. Next-Execution Reconstruction  — models causal consistency by predicting the embedding of the next execution step based on interaction history;
2. Prototype-Guided Enhancement — introduces a learnable prototype prior that provides a stable normality reference under sparse or noisy contexts .
When an anomalous step is detected, MASC triggers a dedicated Correction Agent to revise the output before it propagates downstream. Empirical results on the Who&When benchmark and six MAS frameworks show that MASC, trained entirely on normal trajectories, achieves up to +8.47% AUC-ROC improvement in step-level error detection and 1–3% accuracy gains in end-to-end task performance, all without requiring step-level error labels.

**Strengths:**

1. MASC learns from normal trajectories only, avoiding expensive step-level annotation.
2. Demonstrates consistent improvements across multiple MAS benchmarks, complemented by ablation and score-distribution analyses.
3. The paper is well-structured and easy to follow.

**Weaknesses:**

1. The approach’s effectiveness relies heavily on the representational strength of the underlying encoder; weak embeddings could impair anomaly detection.
2. The Correction Agent’s decision quality and potential for compounding or “secondary” errors are not sufficiently analyzed.
3. Although consistent, the 1–3% end-to-end performance gains might appear modest given the added computation for per-step monitoring.
4. The models used for experiments are not enough, making it not solid.
5. The benchmarks are outdated. More benchs like AIME and MATH should be included.

**Questions:**

* Does the method require computing semantic embeddings at every step during runtime? If so, what is the computational overhead and latency trade-off?
* Are the artificially injected errors in the benchmark distributionally different from naturally emergent reasoning errors in real MAS interactions? Could this limit generalization?
* Why is single-head attention used for prototype updating instead of multi-head or gating variants?
* What is the rationale for combining all-MiniLM-L6-v2 as the encoder and LLaMA-3.1-8B-Instruct as the frozen backbone? How sensitive is MASC to different encoder/backbone choices?
* In Table 3, would the LLM Classifier baseline perform better if a larger or stronger model were used? How does MASC scale relative to such stronger baselines?

---

### Official Review · Reviewer_zEBR · 2025-11-01

**Soundness:** 3
**Presentation:** 2
**Contribution:** 2
**Rating:** 2
**Confidence:** 4

**Summary:**

The paper introduces MASC, an unsupervised framework for detecting and correcting step-level errors in LLM-based multi-agent systems (MAS) using next-execution reconstruction, a prototype for early-step stability, and anomaly-triggered self-correction. Evaluations on the Who&When benchmark and end-to-end tasks claim improvements in detection AUC-ROC and task accuracy.

**Strengths:**

1. MASC can be plugged into existing MAS frameworks without retraining, demonstrated across multiple topologies.
2. In the paper the authors emphasize a rigorous evaluation assessing both the error detector’s standalone performance and its impact on end-to-end task success.

**Weaknesses:**

1. Section 2.2 is unclear about inter-cluster and intra-cluster distances. The text mentions both but only Figure 1a shows intra-cluster distances with/without context. I can see that context helps reduce intra-cluster distance, but there's no visualization or discussion of how inter-cluster distance changes with context, which is equally important for understanding the approach.
2. If normal and abnormal action embeddings have small inter-cluster distance (i.e., they're close together), what guarantees that the detector will actually catch anomalies? There's a real risk that the detector could misclassify correct actions as incorrect, leading to excessive false positives and making the system unreliable.
3. The paper lacks comparisons with SOTA reasoning models like Qwen3, DeepSeek-R1 etc. I strongly encourage the authors to include these models and test them with proper test-time compute, including the "Metacognitive Error Awareness in MAS" approach shown in Figure 1c. Reasoning models have demonstrated ability to self-correct errors when given sufficient compute, making them important baselines.
4. Instead of "Next-Execution Reconstruction", the paper should include a majority voting baseline where you sample the action multiple times with high temperature and apply the same correction pipeline to the history. This is a standard approach that should be compared against.
5. The prototype p isn't well explained: instead of using the real normal embeddings from training data as a steady guide, the method creates p only from the model's own messy guesses (ˆx_t), while tweaking those guesses, p, and attention weights all at once in the same loss function this makes a loop that's shaky and likely to go off track, with no guards from bad early predictions. I would encourage proper intuition for prototype loss.

**Presentation**
1. I would personally encourage the authors to include a brief description about Who&When benchmark for readers unfamiliar with multi-agent systems. Currently, terms like "algorithm generated" and "hand crafted" versions are mentioned without context, making it hard to understand what these variants represent or why they matter just from reading the paragraph in the paper.
2. For the Figure 1, please add a caption explaining what the diagrams represent.
3. The paper is very difficult to follow. There are no clear notations explaining what the LLM and LLM classifier refer to, and although they’re briefly mentioned in the implementation section, it’s still unclear which specific LLM classifier was used the tables don’t make this evident either.

**Questions:**

1. What is the theoretical analysis or intuition for when/why Next-Execution Reconstruction successfully separates normal from anomalous steps? What assumptions are required?
2. Can you provide more details on the "dedicated correction agent"? Is it the same base LLM as the other agents? What is its own failure rate, and have you analyzed scenarios where the correction is also incorrect?
3. What is the actual runtime overhead of MASC compared to vanilla MAS?
4. I would encourage authors to benchmark the setup on much more agentic benchmarks instead of the current ones (GSM8k, MMLU, etc..). The current evaluation setup is not justified for the multi agent setup.

---

### Note · Authors · 2026-01-05

I have read and agree with the venue's withdrawal policy on behalf of myself and my co-authors.